# Identification of digital clinical decision support systems for supporting diagnosis and triage of patients with shoulder disorders: A scoping review protocol

**Cheyenne R. Schamerhorn**[1], **Nathaniel M. Peñas**[2], **Jared R. Fletcher**[1], **Richard Hayman**[3], **Breda H. F. Eubank**[1]*

**1** Department of Health and Physical Education, Faculty of Health, Community, & Education, Mount Royal University, Calgary, Alberta, Canada, **2** Department of Mathematics & Computing, Faculty of Science & Technology, Mount Royal University, Calgary Alberta, Canada, **3** University Library, Mount Royal University, Calgary, Alberta, Canada

* beubank@mtroyal.ca

## Abstract

### Background

Clinical decision support systems (CDSSs) are computerized tools that support clinical decision-making processes. Primary care decision-making is complex and has the potential to influence quality of care provided and patient outcomes. CDSS not only assist providers with clinical decision-making to ensure quality standards are met, reflect evidence-informed practice, and reduce variation in care, but also help patients navigate and receive an appropriate care pathway amidst numerous, often complex, options. Therefore, this scoping review will aim to identify existing CDSSs for supporting primary point-of-care providers, directing patients to appropriate management pathways, and supporting the clinical examination (i.e., medical history-taking and physical examination) process for patients with shoulder disorders. At the primary point-of-care system level, a CDSS for shoulder disorders will improve clinical efficiency and support decision-making.

### Methods

Scoping review methodology and reporting will be conducted according to Arksey and O'Malley's 6-step framework, the Preferred Reporting Items for Systematic Reviews and Meta-Analysis Protocols (PRISMA-P), and the Preferred Reporting Items for Systematic Reviews and Meta-Analysis (PRISMA) extension for Scoping Reviews (PRISMA-ScR) reporting guide. A robust search strategy will be applied across four databases: MEDLINE (Ovid), EMBASE (Ovid), CINAHL (Ebsco), and Scopus (Elsevier). Two blinded reviewers will independently evaluate all titles and corresponding abstracts based on pre-specified inclusion and exclusion criteria. Inter-rater reliability

**Data availability statement:** As this is only a protocol, the data is not yet available. However, upon submission of the final manuscript, the dataset will be publicly available through the MRU Open Access repository: https://library.mtroyal.ca/repository.

**Funding:** - We would like to thank and acknowledge Alberta Innovates which provided CS ($5,625) and NP ($7,500) with a summer research studentship stipend. - https://albertainnovates.ca/funding/summer-research-studentships/ - The funders did NOT play any role in the study design, data collection and analysis, decision to publish, or preparation of the manuscript.

**Competing interests:** The authors have declared that no competing interests exist.

(IRR) agreement will be established during an initial pilot-screening phase against a random selection of 20 records (minimum) until reaching Cohen's Kappa ≥ 0.81. Data extraction will be completed by one reviewer and validated by a second.

## Discussion

An effective and high-quality CDSS that is affordable, easy to use, easily accessible, compatible with existing clinical processes, and generalizable across diverse settings will help to support primary point-of-care providers in diagnosing and managing patients presenting with shoulder disorders, thus improving quality of care for patients.

## Background

Shoulder disorders are one of the most common musculoskeletal (MSK) conditions in the world, in which a significant proportion of the world's population will experience shoulder pain daily, yearly, and throughout a lifetime [1]. Moreover, shoulder disorders are one of the most common reasons for seeking care in the healthcare system [2]. Providing accurate and accessible diagnoses in the publicly-funded medical model, however, is a constant challenge [3]. In the publicly-funded medical model, primary care physicians (i.e., family medicine physicians and general practitioners) are usually the first point of contact for patients in the healthcare system. Primary care physicians, however, often face a time burden during the clinical examination (i.e., medical history-taking, physical examination), sometimes only having two minutes to diagnose a major chief complaint [4]. This burden can lead to creating 'mental shortcuts' to expedite the clinical consult, which may lead to missed questions during the medical history-taking, misdiagnosis, and/or improper referral due to the value that a proper medical history-taking adds to the overall clinical assessment [5]. This can also result in improper treatment and/or referral, which increases the time it takes for the patient to get the proper care they need [5].

Additionally, most primary care physicians have high levels of clinical responsibility and the perceived confidence and competence amongst family medicine physicians and general practitioners in diagnosing and treating shoulder problems remains low [6,7]. There are many uncertainties regarding the clinical management of shoulder disorders, which can result in challenges when deciding on the most appropriate treatment for patients [8]. A recent systematic review of 44 articles reported that regardless of year of study or geographical location (i.e., globally), medical students failed to demonstrate adequate cognitive mastery in the topic of MSK science [9]. Similarly, other primary point-of-care providers such as emergency room physicians, physiotherapists, athletic therapists, and chiropractors, may experience parallel challenges in carrying out an appropriate clinical examination resulting in missed or false diagnoses or inappropriate patient management strategies [10–13].

Primary care providers are restricted by the human body's limited memory and knowledge base, especially with over 200 possible MSK diagnoses to choose from [14].

Additionally, in order to integrate up-to-date evidence-based recommendations in their clinical practice, providers must read a large volume of journals, articles, guidelines, and research outcomes daily [15]. Having a standardized clinical assessment tool to support providers with the physical examination process or in making evidence-based and patient-specific recommendations at the point of care can not only streamline clinical decision-making but also improve quality-of-care for patients presenting with shoulder disorders [16]. Clinical decision support systems (CDSSs) have been developed as a "smart way" to overcome these challenges and can be defined as tools, devices, instruments, questionnaires, algorithms, clinical practice pathways, and treatment models that present health information to assist in patient management decisions [16,17].

CDSSs are computerized systems that use case-based reasoning to assist clinicians in assessing disease status, in making a diagnosis, in selecting appropriate therapy, or in making other clinical decisions [18]. CDSSs also have the potential to reduce the knowledge gap between clinical research and practice, as they allow for an efficient transfer of the latest clinical research [18]. CDSSs have been perceived to be valuable by reducing healthcare costs, streamlining waiting times, increasing appropriateness of referrals, increasing confidence, and improving overall patient care [16]. Over the years, efforts to develop and create CDSSs have increased. CDSSs can be classified as knowledge-based or non-knowledge-based systems [19]. Knowledge-based systems can employ logical rules (IF-THEN statements) drawn from literature, patient-centred protocols, clinical practice guidelines, or expert knowledge used to generate an action or output based on the data entry point [19]. The knowledge-based system will then retrieve the data to evaluate the rule and produce and action or output [19]. Non-knowledge-based systems leverage artificial intelligence, machine learning, or statistical pattern recognition to simulate expert knowledge and require extremely large datasets in the CDSS' development phase [19]. Gross et al. (2016) conducted a scoping review of clinical decision support tools (i.e., non-computerized and computerized) for selecting interventions for patients with disabling musculoskeletal disorders and identified very few computer-based tools (n = 5), of which four were disease scoring questionnaires used to assess health status or categorize patients into risk-groups for care management [16]. Only one CDSS, the Pain Management Advisor, was a knowledge-based CDSS designed to assist physicians in managing patients with complex chronic pain problems based on decision algorithms derived from an expert panel of pain specialists [16]. Unfortunately, Gross et al. (2016) were unable to locate or get further information from the authors of the Pain Management Advisor tool [16].

In the primary care setting, a CDSS that supplements the clinical assessment would help to increase diagnostic accuracy for shoulder disorders as 88% of all diagnoses achieved in primary care and 73% in general medicine are established by the end of the initial history-taking and physical examination [14,20]. A CDSS for assisting in primary care decision-making processes might also help to alleviate the heavy burden primary care providers are faced with such as workforce shortages, burnout, shortened consultation times, and increased administrative burden [21]. Therefore, this study aims to identify CDSSs that support diagnosis and management of patients presenting to primary point-of-care services with shoulder disorders. A secondary objective will be to consolidate the existing evidence for evaluating the: 1) effectiveness of improving patient outcomes; and 2) quality of CDSSs in terms of affordability, accessibility, ease of use, and generalizability across different demographics (e.g., sex, gender, age) and shoulder pathologies.

## Study methods and design

### Study design

This study uses a scoping review methodology and follows Arksey and O'Malley's framework [22]. This protocol was developed following the Preferred Reporting Items for Systematic Reviews and Meta-Analysis Protocols (PRISMA-P) [23], included as S1 Appendix. The Preferred Reporting Items for Systematic Reviews and Meta-Analysis extension for Scoping Reviews (PRISMA-ScR) [24] will be used for reporting results.

**Step 1: identification of the research question.** Inspiration for this scoping review was borrowed from Gross et al., who published a scoping review of clinical decision support tools for selecting interventions for patients with disabling

musculoskeletal disorders [16]. Gross et al. identified both computer and non-computer-based algorithms, care pathways, clinical decision-making rules, and models. The following question will guide our scoping review:

- What knowledge-based or non-knowledge-based CDSSs exist for supporting the diagnosis and management of patients presenting to primary care with shoulder disorders?

### Inclusion criteria

The inclusion criteria are as follows: 1) must describe a CDSS for supporting one or more parts of the clinical assessment such as medical history-taking, physical examination, referrals for diagnostic imaging, referrals to specialist care, and/or treatment; 2) must be designed as either a knowledge-based system and/or non-knowledge-based system; 3) must support shoulder disorders; 4) must be clinician-facing; and 5) available in English due to resource constraints.

### Exclusion criteria

The exclusion criteria will be as follows: 1) CDSSs and other workflow decision-support systems that integrate real-time patient data from hospital visits, electronic health records, medical imaging, and/or laboratory results; 2) CDSSs requiring patient input (e.g., patient-reported health status questionnaires) or interactive testing (e.g., neurocognitive tests, psychological tests, intelligence tests); 3) CDSSs for patient-facing materials such as educational resources, follow-up information, preventive care, and continual monitoring; 4) opinion pieces, editorials, conceptual frameworks, and conference abstracts; 5) reporting only study protocols; and 6) the full-text is not available.

**Step 2: identifying relevant studies.** The search strategy for health literature was informed by the research question and aligned using three primary concepts: CDSS tools, shoulder disorders, and primary health care. Subject terms and other key descriptors were identified using the MeSH thesaurus, reviewing and mining existing research to identify recurring terms, brainstorming and free keyword searching, all tested using MEDLINE. The search string specific to the primary health care concept was informed using an existing expert resource [25], then updated and expanded to meet the needs of the project. Four electronic databases will be searched: MEDLINE (OVID), EMBASE (OVID), CINAHL (Ebsco), and Scopus (Elsevier), selected based on their applicability to health sciences, digital health tools, and medicine. A combination of Medical Subject Headings (MeSH), free keywords, wildcard, proximity, and Boolean operators will be used in the search strings as appropriate, including translation to use database-specific syntax, then expanded for purposes of this study. Search limits will be used to focus primarily on published journal articles and related studies, and limited to English-language publications and human-based studies only. Grey literature will be searched using custom or free-text searching in Google Scholar, OAIster, and medRxiv. The bibliographies of eligible articles will also be hand-searched. Articles and resources found during these searches will be organized using the reference management platform, Covidence [26]. An initial search strategy was created and conducted by a librarian (RH) using Medline (OVID) with current coverage, and is included as S2 Appendix.

**Step 3: selecting studies.** Two reviewers will independently screen titles, abstracts, and full-texts, and select resources based on inclusion/exclusion criteria. Inter-rater reliability (IRR) agreement will be established during an initial pilot-screening phase, where two independent reviewers will apply the inclusion and exclusion criteria against a set of randomly selected records identified during the search process. This pilot screening will be completed within Covidence, which includes IRR data calculations. A minimum of 20 records will be screened, continuing until reaching Cohen's Kappa ≥ 0.81. The first round of screening will focus on titles and abstracts only, using a third reviewer to resolve conflicts. The second round of screening will use the full-text version of only the records that are included during the first phase, also completed using Covidence. Screening phases and results will be reported using a PRISMA flow diagram.

**Step 4: charting the data.** Two reviewers will independently extract relevant data after final agreement of selected resources. Data will be charted into a data extraction worksheet created in Microsoft Excel that has been agreed upon *a*

*priori* by the authorship team (Table 1). One reviewer will lead data charting; however, data points will be independently verified by another member of the authorship team to ensure accuracy and completeness. The following data will be charted: title, authors, year, journal, shoulder pathology, study design, sample size, CDSS type (knowledge-based or non-knowledge based), point in patient care process (e.g., medical history-taking, physical examination, referrals for diagnostic imaging, referrals to specialist care, and/or treatment), CDSS quality indicators, system/user interface (UI) (e.g., mobile, tablet, desktop, graphical, menu-driven, form-based), and effectiveness in improving health outcomes (e.g., synthesis of qualitative or quantitative data/patient reported outcome measures).

**Stage 5: collating, summarizing and reporting the results.** We will provide an overview and consolidate evidence from eligible articles in a numeric, tabular, or chart format. When possible, we will provide summary statistics on our article descriptors and methodological data points. We will also include a descriptive narrative synthesis of findings and provide meaningful recommendations for future direction. We will examine the evidence and assess the quality and effectiveness of CDSSs.

**Step 6: consultation.** During the review process, we will present our findings to stakeholder groups representing a diverse provincial network of clinicians, patients, researchers, policymakers, and health administrators. Participants of this consultation process will be experts in shoulder assessment and management, health research, and policy development. The principal investigator (BE) will facilitate meetings through her network established through ongoing provincial quality improvement work that she is currently involved with (e.g., formerly known as the Bone and Joint Strategic Clinical Network and the Institute for Improved Health Outcomes).

## Ethics and dissemination

Ethics approval was received by the Mount Royal University Human Research Ethics Board (HREB#104043) as Part 1 of the development of a low-fidelity CDSS prototype for the clinical assessment, management, and triage of patients presenting to the health system with shoulder pain. At the end of the study, we will employ traditional reporting approaches (i.e., peer-review manuscripts, presentations) to disseminate this work. We will publish our work in high impact and open access journals to broaden our reach.

## Discussion

The study aligns with the scoping framework outlined by Arksey and O'Malley [22] and will reported using the PRISMA Extension for Scoping Reviews checklist (S1 Appendix) [24] to ensure rigour in our review. The aim of this work will be to identify existing CDSSs for not only directing primary point-of-care providers to appropriate management pathways, but also supporting the clinical examination process (i.e., medical history-taking and physical examination). At the primary point-of-care system level, a CDSS for shoulder disorders would improve clinical efficiency and support decision-making

**Table 1. Data Extraction Form.**

| Article Descriptors | | | | | | Methodology | | CDSS Design | | | | |
|---|---|---|---|---|---|---|---|---|---|---|---|---|
| Article ID | Title | Author(s) | Year | Journal | Shoulder Pathology | Study Design | Sample Size | CDSS Type (Knowledge-based or Non-knowledge-based) | Point in Patient Care Process (e.g., medical history-taking, physical examination, referrals for diagnostic imaging, referrals to specialist care, and/or treatment) | CDSS Quality Indicators (e.g., affordable, accessible, ease of use, generalizable across different demographics and shoulder pathologies) | System/ User Interface (e.g., mobile, tablet, desktop, graphical, menu-driven, form-based) | Effectiveness in improving health outcomes (e.g., synthesis of qualitative or quantitative data/patient reported outcome measures) |
| 101 | | | | | | | | | | | | |

during the clinical examination. This is especially true in rural and remote communities, where providers offer a wide range of services across multiple health conditions, with minimal resources, a heavy workload, in isolation, and with high levels of clinical responsibility and often little MSK training. However, a CDSS must be affordable, easy to use, easily accessible, compatible with existing clinical processes, and generalizable across diverse settings. This would increase clinical uptake and retention as challenges often centre around user acceptance, workflow integration, compatibility with legacy applications, system maturity, and upgrade availability [27]. It is important to note that some providers are concerned about increased CDSS dependence, which limits capacity for independent decision making [27]. However, we anticipate that this review will help to synthesize the available evidence surrounding CDSSs for patients presenting to primary care providers with shoulder disorders.

## Limitations

The inclusion of only English-language articles may limit our scoping review. However, we expect the impact to be minimal given most peer-reviewed articles are published in or translated to English. We will also not be performing a risk of bias assessment as this is a scoping review and narrative synthesis, which may limit the generalizability of any findings.

## Supporting information

**S1 Appendix. PRISMA-ScR Checklist.**
(DOCX)

**S2 Appendix. Search Strategy.**
(DOCX)

## Acknowledgments

We would like to thank and acknowledge Alberta Innovates which provided CS and NP with a summer research studentship and stipend.

## Author contributions

**Conceptualization:** Jared R. Fletcher, Breda H. F. Eubank.

**Data curation:** Cheyenne R. Schamerhorn, Nathaniel M. Penas, Richard Hayman, Breda H. F. Eubank.

**Funding acquisition:** Jared R. Fletcher, Breda H. F. Eubank.

**Methodology:** Richard Hayman.

**Project administration:** Breda H. F. Eubank.

**Validation:** Cheyenne R. Schamerhorn, Nathaniel M. Penas.

**Writing – original draft:** Cheyenne R. Schamerhorn, Breda H. F. Eubank.

**Writing – review & editing:** Cheyenne R. Schamerhorn, Nathaniel M. Penas, Jared R. Fletcher, Richard Hayman, Breda H. F. Eubank.

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
