## [Decision Letter · Decision Letter 0]

Dear Dr. Eubank,

Thank you for submitting your manuscript to PLOS ONE. After careful consideration, we feel that it has merit but does not fully meet PLOS ONE’s publication criteria as it currently stands. Therefore, we invite you to submit a revised version of the manuscript that addresses the points raised during the review process.

We look forward to receiving your revised manuscript.

Kind regards,

Reza Rabiei

Academic Editor

PLOS ONE

Journal Requirements:

Additional Editor Comments (if provided):

Dear Authors,

Thank you for your submission to PLOS One. We have been into the review of the manuscript and based on the reviewers' comments and our assessments, there are minor and major issues with

the manuscript.

Please find the concerns raised by the reviewers, including the inconsistencies remarked and clarifications indicated.

We look forward to receiving your revision and a point-by-point response letter indicating how the comments or the concerns have been handled.

Sincerely,

Reviewers' comments:

Reviewer's Responses to Questions

**Comments to the Author**

1. Does the manuscript provide a valid rationale for the proposed study, with clearly identified and justified research questions?

Reviewer #1: Yes

Reviewer #2: Partly

2. Is the protocol technically sound and planned in a manner that will lead to a meaningful outcome and allow testing the stated hypotheses?

Reviewer #1: Partly

Reviewer #2: Partly

3. Is the methodology feasible and described in sufficient detail to allow the work to be replicable?

Reviewer #1: Yes

Reviewer #2: Yes

4. Have the authors described where all data underlying the findings will be made available when the study is complete?

Reviewer #1: Yes

Reviewer #2: Yes

5. Is the manuscript presented in an intelligible fashion and written in standard English?

Reviewer #1: Yes

Reviewer #2: Yes

You may also provide optional suggestions and comments to authors that they might find helpful in planning their study.

Reviewer #1: The authors describe a pressing need for better support in identifying and appropriately treating musculoskeletal conditions in primary care settings. MSK conditions are relatively common, but complex, and notoriously difficult to accurately diagnose -- especially under the resource-limited conditions of a typical primary care visit. This is an excellent use case for CDSS, but a previous review (2016) failed to identify a viable system. CDSS development has been very active in recent years; a new review has the potential to be highly valuable.

The authors make a clear and compelling case for this scoping review, and I see tremendous potential impact for this work. As it is currently written, however, I see a few inconsistencies that need to be cleared up, and one important issue:

It is unclear why the authors are limiting the scope to knowledge-based CDSS. They discuss potential drawbacks of AI-based CDSS on lines 112-119, but I expected that they would still include and evaluate non-knowledge-based CDSS (and incorporate those potential drawbacks into their evaluation of each CDSS). If there is justification to exclude this entire genre of CDSS at the outset, that should be more clearly explained. Especially given the complexity of the task (200+ potential diagnoses, etc.), there is an obvious potential utility to leveraging machine learning, if it can be done effectively and results checked for bias, etc.

There should be clearer justification for the exclusion criteria. In particular, criteria 1 (see above) and 2, both of which seem like they have the potential to result in more useful CDSS. If there are important constraints (e.g. compatibility with legacy software?) influencing these exclusion criteria, those should be surfaced in the Introduction.

The final sentences of the Discussion make it sound like the intended scope of the tool is shoulder pain specifically, but that is not reflected in the Introduction, proposed methods, or search terms (S2).

MOST IMPORTANT ISSUE: The two guiding research questions on lines 145-149 are not adequately addressed in the proposed methods.

The second question, effectiveness in improving patient care, is likely to be crucial in user acceptance for any new CDSS; de-emphasizing that in favor of measuring "affordability, accessibility, ease of use, compatibility, and generalizability" (line 207) feels like a serious mistake. Table 1 includes a single column for "health impact" -- this should be explained more thoroughly. What kinds of patient outcomes / health impacts will be considered? If this will arise during the course of the review, that should be made clear.

The first guiding question puts emphasis on the initial patient assessment (e.g. medical history-taking), and the justification for doing so is compellingly laid out in the Introduction and Discussion. Unfortunately, data relevant to that are not included in Table 1, unless that is intended to be captured under "CDSS Design". I recommend adding a column capturing what point in the patient care process (e.g. medical history-taking, physical examination, referrals for diagnostic imaging; referrals to specialist care; and/or treatment, line 153-154) each CDSS targets.

Moreover, "affordability, accessibility, ease of use, compatibility, and generalizability" (or similar wording) are mentioned several times as key qualities to look for in CDSS, but I don't see those clearly reflected in either the two guiding research questions or the data to be collected in Table 1 (and, I would argue, something like "effectiveness in improving patient outcomes" should be added to that list).

My overall impression is that the authors have thought very carefully about how to conduct the article search, but the key outcomes to measure are as yet not clearly defined. I think either Table 1 should be substantially revised to better reflect the priorities laid out in the Introduction, or there should be additional guidance included for using Table 1 (i.e. a clear and detailed description for what each cell is intended to capture -- things like "authors" are obvious, but not so for the most important columns, those under Methodology and CDSS Descriptors).

Small typos:

Line 123 "clinicians" is missing a possessive apostrophe (clinicians' or clinician's, depending on whether you intended plural or singular).

Line 204 there is a word missing in the phrase "numeric, tabular, and/or format".

Line 205 "synthesize" should be "synthesis".

Line 230 there is a missing or extra word in the phrase "to identify a knowledge-based CDSS exist"

Line 241-243 is a sentence fragment.

Reviewer #2: The authors describe a scoping review protocol to identify digital clinical decision support systems (CDSS) to support the diagnosis and triage of patients with musculoskeletal (MSK) conditions. The protocol is consistent with best practices for conducting a structured literature review, and is described at an appropriate level of detail.

Despite the overall quality of the presentation, there are several instances of unclear or awkward wording that should be addressed.

It appears based on the article title that the authors intend to formulate a knowledge base that will allow them to assess the state-of-the-art in clinical digital decision support tools for patients with MSK conditions, and ideally to identify a clinical decision support system suitable to their needs. Indeed, this is articulated in the Discussion section where they write, “We anticipate that this review will help to either identify a CDSS that can be used to support point-of-care providers managing patients presenting with shoulder pain or identify similar CDSS where best practices can be borrowed and adapted for our shoulder work.” However, in the last sentence of the Background they write, “This scoping review aims to identify an affordable, easy to use, easily accessible CDSS…”, and in the Discussion, “The aim of this work will be to identify a knowledge-based CDSS exist (?) for not only directing primary point-of-care MSK care providers to appropriate management pathways, but also supporting the clinical examination (i.e. medical history-taking and physical examination) process.” The goal of the scoping review which – again, based on the title – appears to be, first and foremost, to systematically organize information gleaned from multiple sources in the literature, should be clearly articulated and consistent throughout the manuscript. Identifying a specific MSK and/or describing best practices would then be secondary goals of the project.

Additional minor points:

The first sentence of the second paragraph of the Background section reads, “Primary point-of-care providers themselves are restricted by the human body’s limited memory and knowledge base capacity…” This reference to cognitive limitations is well-placed but somewhat awkwardly framed.

In the third paragraph of the Background section the authors state that “Knowledge-based systems are logical rules…” However, this is characterization is not completely accurate because while logical rules are a key component of knowledge-based systems, they include other components such as the user interface and the algorithms that process inputs, operate on the rules, and produce outputs.

The first sentence of the Exclusion criteria section begins with “The following exclusion criteria are as follows,” which should be amended for wording, e.g., “The exclusion criteria are as follows:”

The first sentence of the section “Stage 5: Collating, summarizing, and reporting the results” reads: “We will provide an overview and consolidate evidence from eligible articles in a numeric, tabular, and/or format.” The reference to “and/or format” is ambiguous in this context; this should be expanded upon or rephrased.

**Do you want your identity to be public for this peer review?** For information about this choice, including consent withdrawal, please see our Privacy Policy

Reviewer #1: **Yes: ** Rose M Hartman

Reviewer #2: **Yes: ** Michael E. Bales

---

## [Author Response · Author response to Decision Letter 1]

11 Apr 2025

TOPIC: Review response letter

April 11, 2025

MANUSCRIPT ID: PONE-D-24-41186

Dear Dr Reza Rabiei and Editorial Team:

We thank the reviewers for their helpful review and comments and have detailed the changes made in the submitted manuscript below. As per your instructions, received by email on February 26, 2025, this is our response letter accompanying the submission of manuscript (PONE-D-24-41186) to PLOS One. Below we provide a response to each point raised by the two reviewers from PLOS One, with reference to where changes can be viewed in the submitted manuscript.

Reviewer #1:

The authors describe a pressing need for better support in identifying and appropriately treating musculoskeletal conditions in primary care settings. MSK conditions are relatively common, but complex, and notoriously difficult to accurately diagnose -- especially under the resource-limited conditions of a typical primary care visit. This is an excellent use case for CDSS, but a previous review (2016) failed to identify a viable system. CDSS development has been very active in recent years; a new review has the potential to be highly valuable.

The authors make a clear and compelling case for this scoping review, and I see tremendous potential impact for this work. As it is currently written, however, I see a few inconsistencies that need to be cleared up, and one important issue:

It is unclear why the authors are limiting the scope to knowledge-based CDSS. They discuss potential drawbacks of AI-based CDSS on lines 112-119, but I expected that they would still include and evaluate non-knowledge-based CDSS (and incorporate those potential drawbacks into their evaluation of each CDSS). If there is justification to exclude this entire genre of CDSS at the outset, that should be more clearly explained. Especially given the complexity of the task (200+ potential diagnoses, etc.), there is an obvious potential utility to leveraging machine learning, if it can be done effectively and results checked for bias, etc.

There should be clearer justification for the exclusion criteria. In particular, criteria 1 (see above) and 2, both of which seem like they have the potential to result in more useful CDSS. If there are important constraints (e.g. compatibility with legacy software?) influencing these exclusion criteria, those should be surfaced in the Introduction.

The final sentences of the Discussion make it sound like the intended scope of the tool is shoulder pain specifically, but that is not reflected in the Introduction, proposed methods, or search terms (S2).

Author Response 1:

Thank you for your review of our manuscript. Your comments were insightful. As such, we have revised the inclusion criteria based on your recommendation to include both knowledge-based and non-knowledge-based CDSS. However, since this will impact the volume of our search results, we have refined both the search terms (S2 - Appendix) and inclusion criteria to only “shoulder disorders. We have invited Dr. Richard Hayman, an expert evidence synthesis librarian, to help revise the initial search strategy and added him as a co-author on our manuscript. All co-authors have agreed to this addition (see attached email). This was an oversight on our team in the first submission – which is obviously why there was confusion in our protocol language. The addition of Dr. Richard Hayman will ensure that our process will be clear and further conform to Joanna Briggs Institute and Cochrane Reviews methodologies. We have thus revised the manuscript’s introduction, methods, and discussion section accordingly, which will address Reviewer 1’s concerns above.

Reviewer #1:

MOST IMPORTANT ISSUE: The two guiding research questions on lines 145-149 are not adequately addressed in the proposed methods.

The second question, effectiveness in improving patient care, is likely to be crucial in user acceptance for any new CDSS; de-emphasizing that in favor of measuring "affordability, accessibility, ease of use, compatibility, and generalizability" (line 207) feels like a serious mistake.

Author Response 2:

We agree and have decided that both questions are confusing and do not adequately reflect the scoping review’s primary objective. We have therefore clarified Question 1 to read: “What knowledge-based or non-knowledge-based CDSSs exist for supporting the diagnosis and management of patients presenting to primary care with shoulder disorders?” (lines 16-170) and removed Question 2. Instead, we have replaced Question 2 with the following: “A secondary objective will be to consolidate the existing evidence for evaluating the: 1) effectiveness of improving patient outcomes; and 2) quality of CDSSs in terms of affordability, accessibility, ease of use, and generalizability across different demographics (e.g. sex, gender, age) and shoulder pathologies” (lines 140 – 143).

Reviewer #1:

Table 1 includes a single column for "health impact" -- this should be explained more thoroughly. What kinds of patient outcomes / health impacts will be considered? If this will arise during the course of the review, that should be made clear.

Author Response 2:

We agree that health impact was vague. Our goal is to capture effectiveness in improving health outcomes as suggested. We have revised Table 1 to reflect this change. The table column header now reads “Effectiveness in improving health outcomes (e.g. synthesis of qualitative or quantitative data/patient reported outcome measures)”.

Reviewer #1:

The first guiding question puts emphasis on the initial patient assessment (e.g. medical history-taking), and the justification for doing so is compellingly laid out in the Introduction and Discussion. Unfortunately, data relevant to that are not included in Table 1, unless that is intended to be captured under "CDSS Design". I recommend adding a column capturing what point in the patient care process (e.g. medical history-taking, physical examination, referrals for diagnostic imaging; referrals to specialist care; and/or treatment, line 153-154) each CDSS targets.

Author Response 1:

Thank you for this suggestion. Table 1 has been revised as recommended and now reads “CDSS Point in Patient Care Process (e.g. medical history-taking, physical examination, referrals for diagnostic imaging, referrals to specialist care, and/or treatment)

Reviewer #1:

Moreover, "affordability, accessibility, ease of use, compatibility, and generalizability" (or similar wording) are mentioned several times as key qualities to look for in CDSS, but I don't see those clearly reflected in either the two guiding research questions or the data to be collected in Table 1 (and, I would argue, something like "effectiveness in improving patient outcomes" should be added to that list).

Author Response 1:

Thank you for this suggestion. Table 1 has been revised as recommended under the last column “Effectiveness in improving outcomes (e.g. synthesis of qualitative or quantitative data/patient reported outcome measures)”.

Reviewer #1:

My overall impression is that the authors have thought very carefully about how to conduct the article search, but the key outcomes to measure are as yet not clearly defined. I think either Table 1 should be substantially revised to better reflect the priorities laid out in the Introduction, or there should be additional guidance included for using Table 1 (i.e. a clear and detailed description for what each cell is intended to capture -- things like "authors" are obvious, but not so for the most important columns, those under Methodology and CDSS Descriptors).

Author Response 1:

Thank you for this suggestion and as described above, the manuscript and Table 1 have been revised as you have recommended.

Reviewer #1:

Small typos:

Line 123 "clinicians" is missing a possessive apostrophe (clinicians' or clinician's, depending on whether you intended plural or singular).

Author Response 1:

We have removed this sentence.

Reviewer #1:

Line 204 there is a word missing in the phrase "numeric, tabular, and/or format".

Author Response 1:

We have corrected this sentence.

Reviewer #1:

Line 205 "synthesize" should be "synthesis".

Author Response 1:

We have corrected this sentence.

Reviewer #1:

Line 230 there is a missing or extra word in the phrase "to identify a knowledge-based CDSS exist"

Author Response 1:

We have corrected this sentence.

Reviewer #1:

Line 241-243 is a sentence fragment.

Author Response 1:

We have corrected this sentence.

Reviewer #2:

The authors describe a scoping review protocol to identify digital clinical decision support systems (CDSS) to support the diagnosis and triage of patients with musculoskeletal (MSK) conditions. The protocol is consistent with best practices for conducting a structured literature review, and is described at an appropriate level of detail.

Despite the overall quality of the presentation, there are several instances of unclear or awkward wording that should be addressed.

It appears based on the article title that the authors intend to formulate a knowledge base that will allow them to assess the state-of-the-art in clinical digital decision support tools for patients with MSK conditions, and ideally to identify a clinical decision support system suitable to their needs. Indeed, this is articulated in the Discussion section where they write, “We anticipate that this review will help to either identify a CDSS that can be used to support point-of-care providers managing patients presenting with shoulder pain or identify similar CDSS where best practices can be borrowed and adapted for our shoulder work.” However, in the last sentence of the Background they write, “This scoping review aims to identify an affordable, easy to use, easily accessible CDSS…”, and in the Discussion, “The aim of this work will be to identify a knowledge-based CDSS exist (?) for not only directing primary point-of-care MSK care providers to appropriate management pathways, but also supporting the clinical examination (i.e. medical history-taking and physical examination) process.” The goal of the scoping review which – again, based on the title – appears to be, first and foremost, to systematically organize information gleaned from multiple sources in the literature, should be clearly articulated and consistent throughout the manuscript. Identifying a specific MSK and/or describing best practices would then be secondary goals of the project.

Author Response 2:

Thank you for your review of our manuscript. Your comments were insightful. As mentioned above, we have refined the inclusion criteria to include both knowledge-based and non-knowledge-based CDSS. However, since this will impact the volume of our search results, we have limited both the search terms (S2 - Appendix) and inclusion criteria to only “shoulder disorders. We have invited Dr. Richard Hayman, an expert evidence synthesis librarian, to help revise the initial search strategy and added him as a co-author on our manuscript. All co-authors have agreed to this addition (see attached email). This was an oversight on our team in the first submission – which is obviously why there was confusion in our protocol language. The addition of Dr. Richard Hayman will ensure that our process will be clear and further conform to Joanna Briggs Institute and Cochrane Reviews methodologies. We have thus revised the manuscript’s introduction, methods, and discussion section accordingly, which will address Reviewer 1’s concerns above.

We also agree and have decided that both questions are confusing and do not adequately reflect the scoping review’s primary objective. We have therefore clarified Question 1 to read: “What knowledge-based or non-knowledge-based CDSSs exist for supporting the diagnosis and management of patients presenting to primary care with shoulder disorders?” (lines 16-170) and removed Question 2. Instead, we have replaced Question 2 with the following: “A secondary objective will be to consolidate the existing evidence for evaluating the 1) effectiveness of improving patient outcomes; and 2) quality of CDSSs in terms of affordability, accessibility, ease of use, and generalizability across different demographics (e.g. sex, gender, age) and shoulder pathologies” (lines 140 – 143).

Reviewer #2:

Additional minor points:

The first sentence of the second paragraph of the Background section reads, “Primary point-of-care providers themselves are restricted by the human body’s limited memory and knowledge base capacity…” This reference to cognitive limitations is well-placed but somewhat awkwardly framed.

Author Response 2:

Thank you for your insightful comment. We have revised the statement to read: “Primary care providers are restricted by the human body’s limited memory and knowledge base, especially with over 200 possible MSK diagnoses to choose from.” (lines 90-92)

Reviewer #2:

In the third paragraph of the Background section the authors state that “Knowledge-based systems are logical rules…” However, this is characterization is not completely accurate because while logical rules are a key component of knowledge-based systems, they include other components such as the user interface and the algorithms that process inputs, operate on the rules, and produce outputs.

Author Response 1:

Thank you for your comment. We have revised the statement to read “Knowledge-based systems can employ logical rules (IF-THEN statements) drawn from literature, patient-centred protocols, clinical practice guidelines, or expert knowledge used to generate an action or output based on the data entry point.[19] The knowledge-based system will then retrieve the data to evaluate the rule and produce and action or output.[19]” (lines 110-113)

Reviewer #2:

The first sentence of the Exclusion criteria section begins with “The following exclusion criteria are as follows,” which should be amended for wording, e.g., “The exclusion criteria are as follows:”

Author Response 1:

We agree and have revised as suggested. (line 190)

Reviewer #2:

The first sentence of the section “Stage 5: Collating, summarizing, and reporting the results” reads: “We will provide an overview and consolidate evidence from eligible articles in a numeric, tabular, and/or format.” The reference to “and/or format” is ambiguous in this context; this should be expanded upon or rephrased.

Author Response 1:

We agree and have revised to say: “We will provide an overview and consolidate evidence from eligible articles in a numeric, tabular, or chart format” (lines 259-260)

We again thank the editor and reviewers for their comments, and we hope these revisions are satisfactory.

Respectfully Yours,

Dr. Breda Eubank

---

## [Decision Letter · Decision Letter 1]

Identification of digital clinical decision support systems for supporting diagnosis and triage of patients with shoulder disorders: A scoping review protocol

PONE-D-24-41186R1

Dear Dr. Breda,

We’re pleased to inform you that your manuscript has been judged scientifically suitable for publication and will be formally accepted for publication once it meets all outstanding technical requirements.

Kind regards,

Mansour Abdullah Alshehri

Academic Editor

PLOS ONE

Additional Editor Comments (optional):

After reviewing the authors' response, the addition of Dr. Richard Hayman as a co-author has been appropriately justified in the response to reviewers. There are no serious methodological concerns regarding his inclusion, and his involvement is consistent with the scope and aims of the scoping review.

Reviewers' comments:

Reviewer's Responses to Questions

**Comments to the Author**

1. Does the manuscript provide a valid rationale for the proposed study, with clearly identified and justified research questions?

Reviewer #2: Yes

2. Is the protocol technically sound and planned in a manner that will lead to a meaningful outcome and allow testing the stated hypotheses?

Reviewer #2: Yes

3. Is the methodology feasible and described in sufficient detail to allow the work to be replicable?

Reviewer #2: Yes

4. Have the authors described where all data underlying the findings will be made available when the study is complete?

Reviewer #2: Yes

5. Is the manuscript presented in an intelligible fashion and written in standard English?

Reviewer #2: Yes

You may also provide optional suggestions and comments to authors that they might find helpful in planning their study.

Reviewer #2: The authors have made significant improvements to the manuscript and have adequately addressed this reviewer’s comments.

Inclusion criterion #2 currently reads “must be designed as either a knowledge-based system or non-knowledge-based system”. The authors may consider expanding this to allow for the inclusion of systems that include elements of both knowledge-based and non-knowledge-based systems.

**Do you want your identity to be public for this peer review?** For information about this choice, including consent withdrawal, please see our Privacy Policy

Reviewer #2: **Yes: ** Michael E. Bales

---

## [Editor Report · Acceptance letter]

PONE-D-24-41186R1

PLOS ONE

Dear Dr. Eubank,

I'm pleased to inform you that your manuscript has been deemed suitable for publication in PLOS ONE. Congratulations! Your manuscript is now being handed over to our production team.

Kind regards,

on behalf of

Dr. Mansour Abdullah Alshehri

Academic Editor

PLOS ONE